# C-Type Lectin-like Receptor 2 Expression Is Decreased upon Platelet Activation and Is Lower in Most Tumor Entities Compared to Healthy Controls

**DOI:** 10.3390/cancers15235514

**Published:** 2023-11-22

**Authors:** Mani Etemad, Foteini Christodoulou, Stefanie Uhlig, Jessica C. Hassel, Petra Schrotz-King, Hermann Brenner, Cornelia M. Ulrich, Karen Bieback, Harald Klüter, Peter Bugert

**Affiliations:** 1German Red Cross Blood Service Baden-Württemberg-Hessen, Institute of Transfusion Medicine and Immunology, Medical Faculty Mannheim, Heidelberg University, 69117 Heidelberg, Germany; 2European Center for Angioscience (ECAS), Medical Faculty Mannheim, Heidelberg University, 68167 Mannheim, Germany; 3Flow Core Facility, Medical Faculty Mannheim, Heidelberg University, 68167 Mannheim, Germany; 4Department of Dermatology, National Center for Tumor Diseases, University Hospital Heidelberg, 69120 Heidelberg, Germany; 5Division of Preventive Oncology, German Cancer Research Center (DKFZ), National Center for Tumor Diseases (NCT), 69120 Heidelberg, Germany; 6Division of Clinical Epidemiology and Aging Research, German Cancer Research Center (DKFZ), 69120 Heidelberg, Germany; 7Huntsman Cancer Institute, Department of Population Health Sciences, University of Utah, Salt Lake City, UT 84112, USA

**Keywords:** platelet function, podoplanin, soluble CLEC-2, cancer thrombosis

## Abstract

**Simple Summary:**

Platelets express the C-type lectin-like receptor 2 (CLEC-2) that enables binding to podoplanin (PDPN)-expressing tumor cells and, thereby, promote metastatic spread. An increased level of soluble CLEC-2 was reported in patients with thromboinflammatory and malignant disease, presumably released from activated platelets. In our study, we show that in vitro platelet activation leads to a significant decrease in CLEC-2 on platelets and in the plasma. We also found decreased levels of soluble CLEC-2 in plasma samples of patients with colorectal carcinoma (stages I to IV), breast cancer or melanoma (stages I to III) compared to healthy donors. Interestingly, in patients with glioblastoma, the plasma level of soluble CLEC-2 was significantly higher. We concluded that an increased plasma level of soluble CLEC-2 is not a suitable biomarker of platelet activation and tumor progression in most types of cancer.

**Abstract:**

The C-type lectin-like receptor 2 (CLEC-2) is expressed on platelets and mediates binding to podoplanin (PDPN) on various cell types. The binding to circulating tumor cells (CTCs) leads to platelet activation and promotes metastatic spread. An increased level of soluble CLEC-2 (sCLEC-2), presumably released from activated platelets, was shown in patients with thromboinflammatory and malignant disease. However, the functional role of sCLEC-2 and the mechanism of sCLEC-2 release are not known. In this study, we focused on the effect of platelet activation on CLEC-2 expression and the sCLEC-2 plasma level in patients with cancer. First, citrated blood from healthy volunteer donors (*n* = 20) was used to measure the effect of platelet stimulation by classical agonists and PDPN on aggregation, CLEC-2 expression on platelets with flow cytometry, sCLEC-2 release to the plasma with ELISA and total CLEC-2 expression with Western blot analysis. Second, sCLEC-2 was determined in plasma samples from healthy donors (285) and patients with colorectal carcinoma (CRC; 194), melanoma (160), breast cancer (BC; 99) or glioblastoma (49). PDPN caused a significant increase in the aggregation response induced by classical agonists. ADP or PDPN stimulation of platelets caused a significant decrease in CLEC-2 on platelets and sCLEC-2 in the plasma, whereas total CLEC-2 in platelet lysates remained the same. Thus, the increased plasma level of sCLEC-2 is not a suitable biomarker of platelet activation. In patients with CRC (median 0.9 ng/mL), melanoma (0.9 ng/mL) or BC (0.7 ng/mL), we found significantly lower sCLEC-2 levels (*p* < 0.0001), whereas patients with glioblastoma displayed higher levels (2.6 ng/mL; *p* = 0.0233) compared to healthy controls (2.1 ng/mL). The low sCLEC-2 plasma level observed in most of the tumor entities of our study presumably results from the internalization of sCLEC-2 by activated platelets or binding of sCLEC-2 to CTCs.

## 1. Introduction

Platelets are involved in a variety of physiologic and pathophysiologic processes, including innate immune response, inflammation and atherosclerosis [1,2,3]. Platelets can interact with innate immune cells and secrete cytokines and chemokines [4]. In addition, platelets have a significant impact on the course of many different tumor diseases [5]. Upon contact with cancer cells or circulating tumor cells (CTCs), platelets are activated through a variety of interactions, thereby promoting cancer progression and metastasis [6,7]. The C-type lectin-like receptor 2 (CLEC-2), mainly expressed on megakaryocytes and platelets, is one of the key molecules that enables interaction with other cells, such as innate immune cells and tumor cells [8,9].

CLEC-2 consists of a transmembrane domain, an extracellular lectin-like recognition domain and a short cytosolic tail, harboring a single YxxL sequence, a hem-immunoreceptor tyrosine-based activation motif (hemITAM) [10]. Podoplanin (PDPN) is the most important endogenous ligand of CLEC-2 involved in the cell interaction mechanisms [11]. This glycoprotein with a single transmembrane domain and four platelet aggregation stimulating domains (PLAG) is expressed on many different cells, including lymphatic endothelial cells, kidney podocytes and tumor cells, especially CTCs [12,13,14]. Platelet aggregation induced by PDPN-expressing tumor cells was shown to play a role in tumor growth and metastasis [14,15]. Blocking this interaction led to decreased metastatic herds, better survival and protection against venous thromboembolism (VTE).

Patients with ischemic stroke, acute coronary syndrome or coronary artery disease showed an increased plasma concentration of soluble CLEC-2 (sCLEC-2) [16,17,18]. Similar observations were reported for patients with colorectal cancer (CRC) or glioma [19,20]. It was hypothesized that sCLEC-2 is released upon platelet activation, and therefore, sCLEC-2 could serve as a plasma biomarker of platelet activation [21]. However, the mechanism of sCLEC-2 release is still unclear. In contrast to the release of glycoprotein VI, CLEC-2 is not cleaved by ADAM10/17 [22]. Furthermore, CLEC-2 is not shed or internalized following platelet activation and is maintained on microparticles from activated platelets [23]. An antibody-induced downregulation of CLEC-2 was observed in mice platelets and megakaryocytes, caused through internalization without evidence for ectodomain shedding [24].

To further elucidate the effect of platelet activation and the mechanism of sCLEC-2 release, we first performed in vitro studies on platelets from healthy individuals. We measured the CLEC-2 expression on platelets and the sCLEC-2 plasma concentration after the stimulation of platelets using ADP and PDPN. Second, as an in vivo model for sCLEC-2 release, we measured sCLEC-2 in patients with different solid tumor entities, including colorectal carcinoma, mamma carcinoma, melanoma and glioblastoma.

## 2. Materials and Methods

### 2.1. Blood Samples

Citrated blood samples (with 3.2% citrate) from 20 healthy volunteer donors (8 male, 12 female; mean age 47.6 ± 8.9 y) were used for aggregometry and the in vitro studies with flow cytometry, ELISA and Western blot analysis. At the time of blood collection, the donors had no medication affecting platelet function in the past 10 days. Platelet-rich plasma (PRP) was prepared from citrated blood with centrifugation for 15 min at 200 g in a swingout rotor within three hours after blood collection. For the in vitro platelet stimulation experiments, 500 μL PRP from healthy donors was incubated without an agonist or with 5 μM ADP (möLab GmbH, Langenfeld, Germany) or PDPN (1 μg and 5 μg recombinant human glycosylated PDPN protein; Sino Biological Inc.; Biozol GmbH, Eiching, Germany) at room temperature for 30 min with slight agitation. The PRP samples were then directly subjected to flow cytometry, ELISA and Western blot analysis.

For the determination of the sCLEC-2 plasma level, a standardized procedure was used to obtain plasma from healthy control donors and patients with cancer. Briefly, plasma was obtained from EDTA blood samples (with 1.6 mg K3 EDTA per mL blood) after centrifugation at 1500 g for 10 min. within 24 h after blood sampling and stored at −80 °C until use for ELISA. The plasma samples of the 285 randomized healthy blood donors were already used in a previous study [25]. The plasma samples from patients with colorectal cancer (CRC; *n* = 194), melanoma (160) and breast cancer (BC; 99) were provided by the National Center for Tumor Diseases (NCT) Heidelberg. Samples from 49 patients with glioblastoma were collected at the Center for Neurooncology, Comprehensive Cancer Center, Tübingen. All donors and patients gave informed consent to provide blood samples for research purposes. The study protocols were in accordance with the Declaration of Helsinki and approved by the ethical committee of the Medical Faculty Mannheim of the Heidelberg University (ethical approval codes: 2016-615N-MA, 2016-657N-MA).

### 2.2. Aggregometry

The platelet aggregation response was measured using light transmission aggregometry (LTA) in the PAP-8 system (möLab GmbH). In pilot experiments, we used PDPN in different concentrations and incubation times for the stimulation of platelets, which did not lead to a platelet aggregation response. Therefore, PDPN was used as a co-agonist together with the classical agonists ADP, arachidonic acid (AA) and collagen (COL) to stimulate platelets and determine the aggregation response for 10 min. As given in the results, ADP, AA and COL were used at the standard concentration and 5 times lower than the standard concentration. The primary aggregation (PA%) values were used for data analysis and statistical evaluation.

### 2.3. Flow Cytometry

Flow cytometry was used to measure the expression of CLEC-2 and the activation marker CD62P (P-selectin) on platelets of healthy donors. The following antibodies were used: monoclonal anti-CD62P (PE-conjugated, clone AK-4, IgG1; BD Biosciences, Heidelberg, Germany) and monoclonal anti-CLEC-2 (FITC-conjugated, clone 17D9, IgG2b; Thermo Fisher Scientific, Darmstadt, Germany). For in vitro platelet stimulation, we used ADP and recombinant human glycosylated PDPN protein. Unstimulated as well as ADP- and PDPN-stimulated PRP was assessed unstained and double-stained for CLEC-2 (FITC) and CD62P (PE) by using a FACS-Canto II system (BD Biosciences). All events detected in the stained samples were gated according to the side and forward scatter and further analyzed for mean fluorescence intensity (MFI) in each fluorescence channel. The results were evaluated and processed for graphical presentation with the FlowJo software version 9 (FlowJo, LLC; Ashland, OR, USA).

### 2.4. ELISA for sCLEC-2

A commercial ELISA kit (RayBiotech; Hoelzel Diagnostika GmbH, Cologne, Germany) was used to measure the sCLEC-2 in the plasma samples from the in vitro stimulation experiments and in the plasma samples from the healthy blood donors and the tumor patients. Unstimulated as well as ADP- and PDPN-stimulated PRP was centrifuged for 10 min at 1000 g. The plasma in the supernatant was transferred to another tube and further processed for ELISA according to the standard protocol. The platelet pellet was further processed for Western blot analysis. The sCLEC-2 concentration was measured in all plasma samples in duplicate according to the standard protocol of the ELISA kit. Standardization of the optical density (OD) measured at 450 nm was achieved by using a dilution series of the recombinant CLEC-2 protein provided with the kit. The linear range of the standard curve was 0.1–48 ng/mL sCLEC 2 with a significant correlation (r^2^ = 0.9984).

### 2.5. Western Blot

The pellets of the unstimulated as well as ADP- and PDPN-stimulated platelets were resuspended in cold lysis buffer (RIPA buffer including protease inhibitor cocktail; Santa Cruz Biotechnology, Inc., Heidelberg, Germany) followed by incubation on ice for 30 min. After shock freezing in liquid nitrogen followed by thawing, ultrasound treatment and centrifugation for 10 min at 12,000 g, the supernatant (platelet lysate) was transferred to another tube. The total protein concentration of the lysates was determined by using a Bradford protein assay (Quick Start™ Bradford Protein Assay; Bio-Rad Laboratories GmbH, Feldkirchen, Germany). The sample volume corresponding to 25 μg total protein was calculated and mixed with electrophoresis sample buffer (Santa Cruz Biotechnology Inc., Dallas, TX, USA). After heat denaturation for 5 min at 100 °C, the samples were loaded on 8.75% polyacrylamid gels (PAGE) and electrophoresed for 50 min at 25 V/cm. The separated proteins were blotted onto nitrocellulose membranes (Bio-Rad) with electric field transfer. After incubation of the blots in blocking solution (UltraCruz^®^ Blocking Reagent; Santa Cruz Biotechnology Inc.) for 1 h, the primary antibody (goat anti-human CLEC1B polyclonal antibody; Invitrogen, Darmstadt, Germany) was added and incubated overnight at 4 °C with constant agitation. Then, the blots were washed and incubated with the secondary antibody (mouse anti-goat HRP-conjugated; Santa Cruz Biotechnology Inc.) for one hour at room temperature. After repeated washing steps, we added Luminol and peroxide solution and measured our blots using the Epi Chemi II Darkroom (UVP laboratory products, Upland, CA, USA). For relative quantification of the CLEC-2 protein band, the blots were stained for beta-actin (monoclonal anti-actin antibody, HRP-conjugated; Santa Cruz Biotechnology Inc.) as the reference protein. Quantitative analysis of the protein bands was accomplished with the ImageJ program.

### 2.6. Statistical Analysis

*t*-tests and further statistical analyses were performed using Excel version 12 (Microsoft Corp., Redmond, WA, USA) and PRISM version 8 (GraphPad Software, Boston MA, USA). *p* values < 0.05 were considered as statistically significant.

## 3. Results

First, we investigated the effect of PDPN on the platelet aggregation response of heathy donors. Second, ADP and PDPN were used as agonists for in vitro stimulation of platelets from healthy donors to analyze the expression of CLEC-2 on platelets, in plasma and in platelet lysates. The third part of our investigation included the measurement of sCLEC-2 in plasma samples from healthy controls and patients with different tumor entities.

### 3.1. PDPN Is a Co-Agonist for Platelet Aggregation

Platelets from 20 healthy volunteer donors were stimulated for aggregation using the classical agonists ADP, AA and COL at the corresponding standard concentration for LTA and five times lower concentration. To determine the effect of PDPN, we applied 1 μg and 5 μg of the recombinant protein to the samples with 225 μL PRP. PDPN on its own did not induce a platelet aggregation response. Only the classical agonists induced platelet aggregation (Figure 1). As expected, the aggregation response was dose-dependently reduced when using the five-fold lower agonist concentration. The co-stimulation with 5 μg PDPN caused a significant increase in the primary aggregation induced by all the classical agonists, while 1 μg showed no effect (Table 1).

### 3.2. The CLEC-2 Expression Is Decreased upon Platelet Stimulation

The expression of CLEC-2 and the activation marker CD62P was determined on platelets without stimulation and after stimulation with ADP and PDPN. The results from flow cytometry revealed a significant decrease in the CLEC-2 expression upon ADP and PDPN stimulation of the platelets (Figure 2). As expected, the CD62P expression was increased upon ADP and PDPN stimulation.

In order to prove whether the reduced CLEC-2 expression upon platelet activation was caused by shedding of CLEC-2 and release to the plasma, sCLEC-2 was measured in the supernatant of the ADP- and PDPN-stimulated PRP. Platelet activation was associated with a significant decrease in sCLEC-2 in the plasma (Figure 3a), whereas the relative quantification of total CLEC-2 in the platelet lysates indicated a comparable protein concentration (Figure 3b).

These data suggest that shedding and degradation of CLEC-2 upon platelet activation are rather unlikely (Table 2). Given that the overall expression levels do not change, internalization of CLEC-2 in platelets could be the cause.

### 3.3. The Plasma Level of sCLEC-2 Is Lower in Tumor Patients Than in Healthy Controls

Previous studies reported increased sCLEC-2 plasma levels in patients with inflammatory and malignant disease, and it was hypothesized that sCLEC-2 is released upon platelet activation. However, our in vitro studies indicated that platelet activation is not associated with increased but rather decreased CLEC-2 expression and sCLEC-2 level in the plasma. To further investigate this in vivo, we measured sCLEC-2 in the plasma of healthy donors and patients with different solid tumors. We included 285 healthy controls (HC), 194 patients with colorectal cancer (CRC), 160 patients with melanoma, 99 patients with breast cancer (BC) and 49 patients with glioblastoma. Table 3 includes the demographic data, indicating significant differences regarding age and gender between HC and patients.

Compared to the healthy controls (HC), we found significantly lower (*p* < 0.0001) sCLEC-2 plasma levels in patients with CRC, melanoma and BC (Figure 4). Interestingly, the sCLEC-2 level in patients with glioblastoma was significantly higher (*p* = 0.0233). A correlation to the tumor stages was assessed for CRC and melanoma. All CRC stages I to IV showed significantly lower sCLEC-2 levels than HC (Figure 5, Table 4). For the melanoma stages I to III, we also found significantly lower sCLEC-2 levels, whereas in patients with stage IV tumors, the sCLEC-2 level was comparable with HC.

In summary, the in vivo data from patients revealed significantly lower sCLEC-2 plasma levels compared to heathy controls for all tumor entities analyzed except melanoma stage IV and glioblastoma. The latter displayed higher sCLEC-2 levels.

## 4. Discussion

It has been clearly demonstrated that the binding of PDPN to CLEC-2 leads to an activation of platelets. However, with regard to platelet aggregation, PDPN was only a weak agonist that enhanced stimulation by other agonists, such as ADP, AA or collagen. In contrast to other platelet receptors, such as CD62P or GPVI, we showed that the CLEC-2 expression on platelets and the sCLEC-2 level in the plasma are decreased upon platelet activation. Thus, shedding of CLEC-2 upon platelet activation is rather excluded. We hypothesize that platelet activation causes the internalization of CLEC-2, including the uptake of sCLEC-2 from the plasma. The reported antibody-induced internalization of CLEC-2 in mouse platelets could represent a similar mechanism [24]. However, the findings from our in vitro studies are contradictory to the reported increased sCLEC-2 plasma levels in patients with thromboinflammatory diseases that are associated with increased platelet activation [16,17,18,19,20,21,22]. These studies proposed sCLEC-2 as a plasma biomarker for platelet activation [21]. It remains unclear whether the difference in the in vitro conditions in our studies and the in vivo conditions or the pathologic condition accounts for the contradictory results.

Increased sCLEC-2 plasma levels were also reported for colorectal cancer and glioma [19,20]. The CLEC-2/PDPN-mediated binding of platelets to CTCs further promotes the crucial role of CLEC-2 in the progression and metastasizing of tumors [7,12,13]. The CLEC-2/PDPN axis was discussed as a promising drug target for cancer treatment [26,27]. Antibody-mediated inhibition of the interaction between tumor PDPN and platelet CLEC-2 blocked the growth and pulmonary metastasis of human malignant melanoma [28]. As shown in a previous study, the decrease in CLEC-2 expression in gastric cancer tissues and cell lines led to a lower invasiveness of the cells [29]. These findings indicated a tumor-inhibitory role for cancer cell resident CLEC-2. Notably, these experiments were conducted under normal platelet count conditions in mice, suggesting that platelet-derived CLEC-2 is dispensable in this cancer type.

In contrast to previous reports, we observed significantly lower sCLEC-2 plasma levels in patients with cancer compared to healthy donors. Zang et al. showed higher sCLEC-2 levels in patients with CRC compared to healthy controls [19]. Moreover, the patients with liver metastases displayed higher CLEC-2 levels than patients without metastases. In our study, the sCLEC-2 level in patients with CRC at stages I to IV was significantly lower than in healthy controls. The same was true for melanoma stages I to III. The 56 patients with stage IV melanoma displayed sCLEC-2 levels comparable with healthy donors but significantly higher compared to stages I to III melanoma. Patients with melanoma have a considerably higher risk of suffering from VTE than patients with CRC or BC [30,31,32]. In stage IV melanoma, a prevalence of VTE of 25% was observed [30]. Glioblastoma was the only tumor entity in our study that was associated with higher sCLEC-2 levels and confirmed the observation of previous reports [20]. A correlation between sCLEC-2 levels and PDPN expression was reported in patients with high-grade gliomas [20]. Interestingly, high PDPN expression in primary brain tumors induces platelet aggregation and is associated with increased risk of VTE [33]. As shown in a mouse model, blocking PDPN specifically on glioblastoma cells could represent a novel strategy to prevent platelet aggregation and, thereby, reduce the risk of VTE in glioma [34]. However, tumor progression occurs independently of PDPN, and the blocking of PDPN does not represent a promising anti-cancer therapeutic approach [35]. Taking the observations from our in vitro platelet stimulation experiments into account, the significantly lower levels of sCLEC-2 in the plasma of patients with cancer may result from the internalization of sCLEC-2 by activated platelets. The binding of sCLEC-2 to PDPN on CTCs should also be considered. In this concept, platelet CLEC-2 is shed, and sCLEC-2 is trapped in the microenvironment of primary or metastatic tumors, leading to reduced sCLEC-2 levels in the blood plasma of patients with cancer. The trapping mechanism, probably, does not exist in glioblastoma. In a follow-up study, we will address the mechanism by using different cancer cell lines, including CRC, BC and glioblastoma, to test the binding or uptake of sCLEC-2.

In our study, the mean age of the patients was significantly higher compared to that of HC. However, we found no association of age with the sCLEC-2 plasma levels, neither in healthy individuals nor in patients with cancer. The significant difference between HC and patients with regard to the male:female ratio did also not explain the differences in the sCLEC-2 plasma levels. In general, the sCLEC-2 plasma levels in males and females are very similar. Another potential explanation of the different values measured in the samples from different sources is the processing and storage of the samples. In our study, a standardized sampling procedure ensured very similar sample quality and comparability of the results. Another aspect regarding the differences between the HC and patients is given by the individual genetic background and ethnicity. However, the genetic data of the patients with cancer in the present study were not available, but we assume that the ethnicity of the HC and the patients with cancer is very similar. In a previous study, we reported a correlation of haplotypes of the *CLEC1B* gene encoding CLEC-2 with the sCLEC-2 level in healthy individuals [25]. The haplotypes were defined by seven single nucleotide variations (SNVs: rs10505743, rs11053538, rs4764178, rs76016091, rs2273986, rs2273987, rs521040), and the most frequent haplotype was associated with significantly lower sCLEC-2 levels. According to the gnomAD database (https://gnomad.broadinstitute.org; accessed on 5 November 2023), the minor allele frequency of some SNVs is different in the European, African and Asian populations (e.g., rs4764178 and rs2273986 are more frequent in Europeans than in Asians, whereas rs2273987 and rs521040 are more frequent in Asians than in Europeans).

Cancer-associated VTE is associated with an increased risk of mortality independent of the cancer characteristics. Nowadays, the use of direct oral anticoagulants (DOACs) is the most effective therapy for managing VTE in the general population and is also recommended for patients with cancer (for review, see [36]). Anti-platelet therapy using aspirin can reduce both the long-term risk of cancer and the risk of cancer metastasis for some solid tumors [36,37,38]. The ADD-ASPIRIN trial is a phase III, double-blind, placebo-controlled, randomized trial that investigates whether regular aspirin use after standard therapy prevents recurrence and prolongs survival in participants with non-metastatic CRC, BC, gastro-esophageal cancer or prostate cancer [39]. A better understanding of the mechanisms of how platelets promote tumor growth and progression is a prerequisite for the development of novel anti-cancer therapies.

## 5. Conclusions

Based on our in vitro studies on platelets from healthy individuals, we showed that the CLEC-2 expression on platelets and the sCLEC-2 level in the plasma are decreased upon platelet activation. The uptake and internalization of the receptor is the proposed mechanism. Therefore, we conclude that an increased plasma level of sCLEC-2 is not a suitable biomarker of platelet activation. We also found decreased levels of sCLEC-2 in the plasma samples of patients with CRC, BC and melanoma stages I to III compared to healthy controls. Glioblastoma was the only tumor entity in our study with significantly higher sCLEC-2 levels. CLEC-2 and the ligand PDPN might play a significant role in the development and progression of glioblastoma as already hypothesized by other studies. The low sCLEC-2 plasma level observed in most of the tumor entities of our study presumably results from the internalization of sCLEC-2 by activated platelets or the binding of sCLEC-2 to CTCs. 

## Figures and Tables

**Figure 1 cancers-15-05514-f001:**
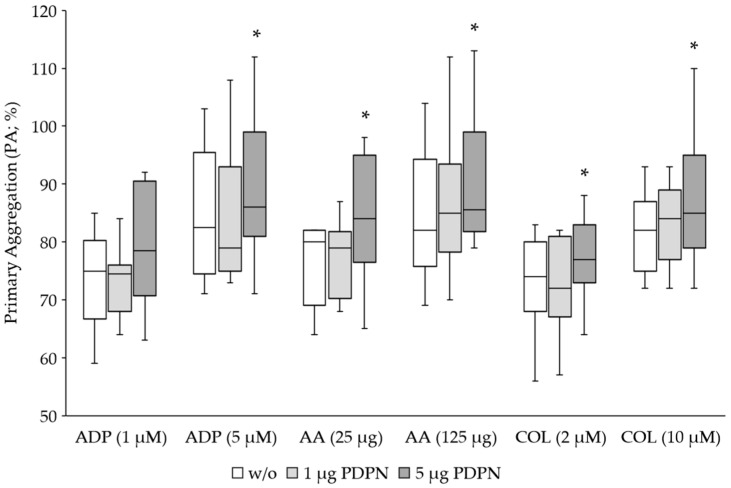
Aggregation response of healthy donors (*n* = 20) upon stimulation of platelets with the classical agonists ADP, AA and COL and PDPN as the co-agonist. The primary aggregation (PA) is given in % and corresponded to the maximum aggregation in all measurements. Box-whisker plots show the 25th–75th percentile (boxes) with the median indicated by the line inside each box, and the 1.5 interquartile range by vertical lines (whiskers). The use of 5 μg PDPN caused a significant (* *p* < 0.05) increase in the aggregation response (*n* = 10).

**Figure 2 cancers-15-05514-f002:**
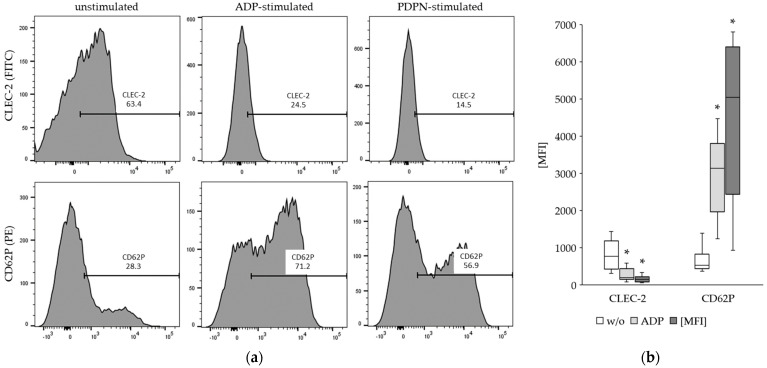
Results from flow cytometry for the detection of CLEC-2 and CD62P expression on platelets: (**a**) Representative result from one donor, indicating a decrease in CLEC-2 and increase in CD62P expression upon ADP and PDPN stimulation; (**b**) Summary of the results from 20 healthy donors. The decrease in CLEC-2 expression and the increase in CD62P expression were significant (*; *p* < 0.001) for ADP and PDPN stimulation. For further decription of the box-whisker plots see legend to Figure 1.

**Figure 3 cancers-15-05514-f003:**
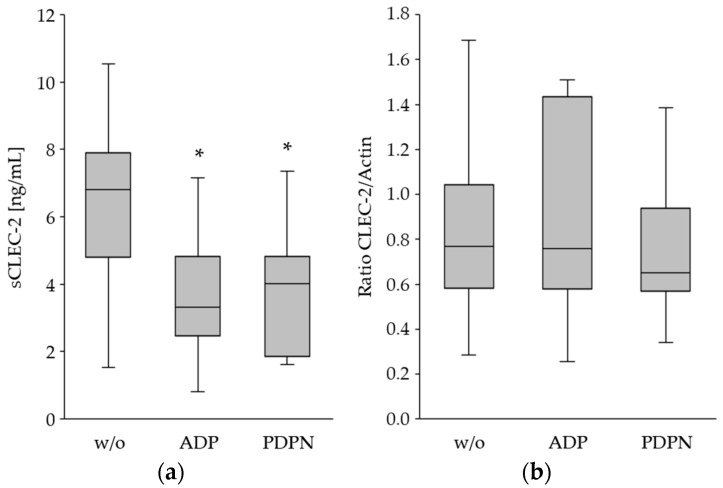
Measurement of sCLEC-2 in plasma (**a**) and total CLEC-2 in platelet lysates (**b**) after ADP and PDPN stimulation. (**a**) sCLEC-2 levels in unstimulated (w/o) and ADP- and-PDPN stimulated PRP by using ELISA; (**b**) Relative quantification of total CLEC-2 in platelet lysates without activation and ADP and PDPN stimulation with Western blot analysis and actin as the reference protein. The decrease in sCLEC-2 upon stimulation was significant (*; *p* < 0.001), whereas the total CLEC-2 protein concentration was comparable. For further decription of the box-whisker plots see legend to Figure 1.

**Figure 4 cancers-15-05514-f004:**
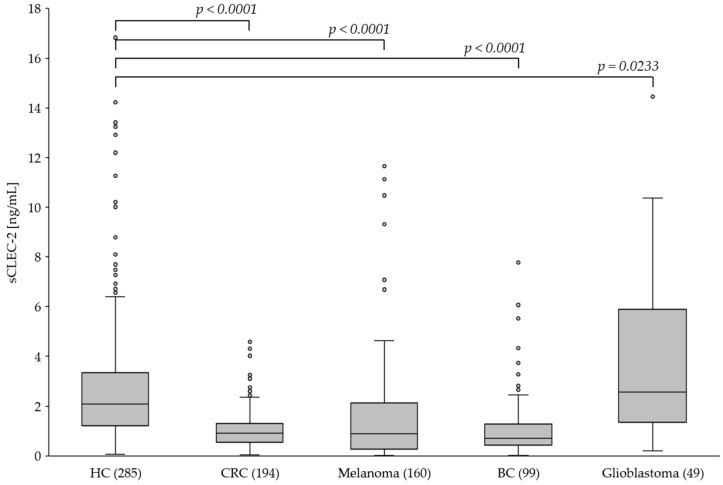
The plasma level of sCLEC-2 in healthy controls (HC; *n* = 285) and patients with colorectal cancer (CRC; *n* = 194), melanoma (*n* = 160), breast cancer (BC; *n* = 99) and glioblastoma (*n* = 49). For further decription of the box-whisker ploets see legend to Figure 1. Circles indicate outliers.

**Figure 5 cancers-15-05514-f005:**
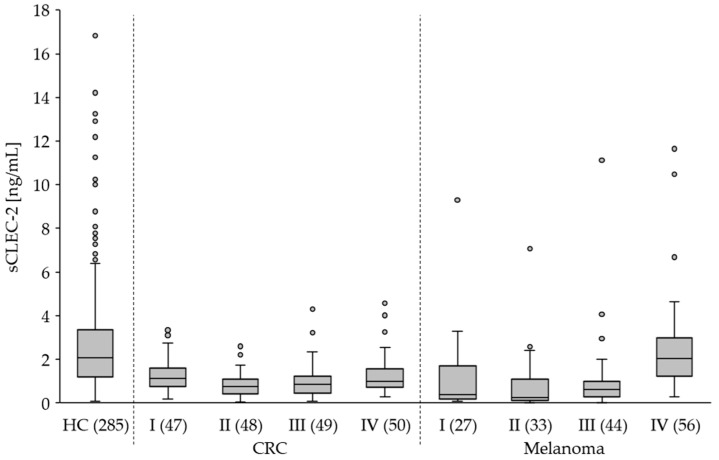
The plasma level of sCLEC-2 in patients with CRC and melanoma stages I to VI compared to healthy controls (HC). N in brackets. Box plots show the 25th–75th percentile with the median indicated by the line inside each box. Whiskers represent the 1.5 interquartile range (circles indicate outliers).

**Table 1 cancers-15-05514-t001:** Statistical evaluation of the primary aggregation (PA; %).

Agonist	No PDPN	1 μg PDPN	5 μg PDPN	*p*-Value ^1^	*p*-Value ^2^
ADP (1 μM)	74 ± 8	73 ± 6	79 ± 10	0.7242	0.0829
ADP (5 μM)	85 ± 11	84 ± 12	89 ± 11	0.7905	**0.0253**
AA (25 μg)	76 ± 7	78 ± 6	84 ± 10	0.2642	**0.0072**
AA (125 μg)	85 ± 11	87 ± 13	91 ± 12	0.1578	**0.0110**
COL (2 μM)	72 ± 8	73 ± 7	77 ± 7	0.9671	**0.0005**
COL (10 μM)	82 ± 6	83 ± 7	88 ± 10	0.1236	**0.0062**

^1^ paired *t*-test for the significance of the difference between no PDPN and 1 μg PDPN; ^2^ paired *t*-test for the significance of the difference between no PDPN and 5 μg PDPN; bold numbers indicate significant *p*-values < 0.05.

**Table 2 cancers-15-05514-t002:** Statistic evaluation of the platelet in vitro studies.

	No Agonist	ADP	PDPN	*p*-Value ^1^	*p*-Value ^2^
CLEC2 (MFI)	792 ± 373	259 ± 154	158 ± 77	**<0.0001**	**<0.0001**
CD62P (MFI)	661 ± 292	2948 ± 1021	4496 ± 1957	**<0.0001**	**<0.0001**
sCLEC-2 (ng/mL)	6.3 ± 2.6	3.6 ± 1.7	3.8 ± 1.7	**0.0002**	**0.0006**
Total CLEC-2 (ratio) *	0.83 ± 0.41	0.87 ± 0.42	0.74 ± 0.31	0.8518	0.6785

^1^ paired *t*-test for the significance of the difference between no agonist and ADP stimulation; ^2^ paired *t*-test for the significance of the difference between no agonist and PDPN stimulation; * the ratio of the CLEC-2 and the actin protein bands was used as a measure of the total CLEC-2 content; bold numbers indicate significant *p*-values < 0.05.

**Table 3 cancers-15-05514-t003:** Demographic information about the study cohort including healthy controls and patients with cancer.

	Age, y (Mean ± SD)	Gender, *n* (Male:Female)	*p*-Value ^1^	*p*-Value ^2^
HC (*n* = 285)	44.3 ± 13.5	208:77		
CRC (*n* = 194)	69.7 ± 18.9	113:81	<0.0001	0.0011
Melanoma (*n* = 160)	59.5 ± 15.2	88:72	<0.0001	0.0002
BC (*n* = 99)	62.6 ± 17.5	0:99	<0.0001	-
Glioblastoma (*n* = 49)	62.0 ± 13.9	26:23	<0.0001	0.0082

^1^ unpaired *t*-test for the significance of the age difference between HC and patient group; ^2^ Chi^2^ test for the difference in male:female ratio between HC and patient group.

**Table 4 cancers-15-05514-t004:** Statistic evaluation of the sCLEC-2 levels in healthy controls and patients with different tumor entities and stages.

Study Group (*n*)	sCLEC-2 (ng/mL) (Median)	sCLEC-2 (ng/mL) (Mean ± SD)	*p*-Value ^1^
HC (285)	2.1	2.8 ± 2.6	
CRC (194) I (47) II (48) III (49) IV (50)	0.9 1.1 0.8 0.9 1.0	1.1 ± 0.9 1.3 ± 0.8 0.8 ± 0.6 1.0 ± 0.8 1.3 ± 1.0	**<0.0001** **0.0002** **<0.0001** **<0.0001** **0.0001**
Melanoma (160) I (27) II (33) III (44) IV (56)	0.9 0.4 0.2 0.6 2.0	1.5 ± 2.0 1.1 ± 1.9 0.8 ± 1.3 1.0 ± 1.7 2.5 ± 2.1	**<0.0001** **0.0016** **<0.0001** **<0.0001** 0.4319
BC (99)	0.7	1.2 ± 1.3	**<0.0001**
Glioblastoma (49)	2.6	3.7 ± 3.3	**0.0233**

^1^ independent *t*-test for the significance of the difference between HC and the patient group; values are given for all CRC and melanoma and for the stage subgroups I to IV; bold numbers indicate significant *p*-values < 0.05.

## Data Availability

All data presented in this study are available in this article.

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
