# Peer review of "C-Type Lectin-like Receptor 2 Expression Is Decreased upon Platelet Activation and Is Lower in Most Tumor Entities Compared to Healthy Controls"

_cancers, 2023, doi:10.3390/cancers15235514_

Round 1

Reviewer 1 Report

Comments and Suggestions for Authors

The in vitro experiments and results of platelet activation with ADP and podoplanin to measure CLEC-2 expression on and in platelets and the dynamic changes in plasma are very interesting. It would be good to add information about the percentage of sodium citrate in the used blood tubes. It would also be good to state the age and gender distribution of these healthy subjects.

Unfortunately, the clinical data for measuring the CLEC2 concentration has some weaknesses.

A demographic table with information on the age and gender distribution of the cancer patients (n = 388) and the healthy controls (n = 285) is missing. Were the age and gender distribution of the healthy controls matched to the respective cancer patient groups? The plasma samples of the healthy controls were obtained from EDTA blood. However, it is unclear from the description of the materials and methods whether the plasma samples from the cancer patients were also obtained from EDTA blood. Information about the centrifugation conditions for obtaining plasma samples from healthy controls and cancer patients is also missing.

The significantly reduced amount of CLEC-2 in the plasma of CRC, melanoma or BC patients could also be because the controls were not age and gender-matched, and/or different blood tubes and centrifugation conditions were used to generate plasma samples.

Author Response

Reviewer: 1

The in vitro experiments and results of platelet activation with ADP and podoplanin to measure CLEC-2 expression on and in platelets and the dynamic changes in plasma are very interesting. It would be good to add information about the percentage of sodium citrate in the used blood tubes. It would also be good to state the age and gender distribution of these healthy subjects.

We added the corresponding information to the Methods section:

Citrated blood samples (with 3.2 % citrate) from 20 healthy volunteer donors (8 male,    12 female; mean age 47.6 ± 8.9 y) were used for aggregometry and the in vitro studies       with flow cytometry, ELISA and Western blot analysis.

A demographic table with information on the age and gender distribution of the cancer patients (n = 388) and the healthy controls (n = 285) is missing. Were the age and gender distribution of the healthy controls matched to the respective cancer patient groups? The plasma samples of the healthy controls were obtained from EDTA blood. However, it is unclear from the description of the materials and methods whether the plasma samples from the cancer patients were also obtained from EDTA blood. Information about the centrifugation conditions for obtaining plasma samples from healthy controls and cancer patients is also missing.

The significantly reduced amount of CLEC-2 in the plasma of CRC, melanoma or BC patients could also be because the controls were not age and gender-matched, and/or different blood tubes and centrifugation conditions were used to generate plasma samples.

The missing infomation is added to the Methods section. 

For the determination of the sCLEC-2 plasma level, a standardized procedure was used to obtain plasma from healthy control donors and cancer patients. Briefly, plasma was obtained from EDTA blood samples (with 1.6 mg K3 EDTA per mL blood) after centrifugation at 1,500 g for 10 min. within 24 h after blood sampling and stored at -80 °C until use for ELISA.

In the Results section we added one paragraph and the new Table 3 including demographic data:

We included 285 healthy controls (HC), 194 colorectal cancer (CRC), 160 melanoma, 99 breast cancer (BC) and 49 glioblastoma patients. Table 3 include demographic data indicating significant differences regarding age and gender between HC and patients.

This point was further discussed in an additional paragraph in the Discussion section:

In our study, the mean age of the patients was significantly higher compared to HC. However, we found no association of age with the sCLEC-2 plasma levels neither in healthy individuals nor in cancer patients. The significant difference between HC and patients with regard to the male:female ratio did also not explain the differences in sCLEC-2 plasma levels. In general, sCLEC-2 plasma levels in males and females are very similar. Another potential explanation of different values measured in samples from different sources is the processing and storage of samples. In our study a stand-ardized sampling procedure ensured a very similar samples quality and comparability of the results.

Reviewer 2 Report

Comments and Suggestions for Authors

In the presented manuscript entitled "C-type Lectin-like Receptor 2 Expression is Decreased Upon Platelet Activation and is Lower in Most Tumor Entities Compared to Healthy Controls," Etemad et al. measured the soluble form of CLEC-2 (sCLEC-2) in the blood plasma of different cancer patients, including those with colorectal adenocarcinoma (CRC), melanoma, breast cancer (BC), and glioblastoma. In contrast to previously published results, sCLEC-2 levels were decreased in patients with CRC, with elevated levels of sCLEC-2 observed only in patients with glioblastoma. Furthermore, the authors demonstrated that podoplanin alone could induce platelet degranulation, exposing P-Selectin (CD62P) as a marker of alpha-granule secretion on the platelet surface, but it does not induce platelet aggregation. The protein expression of CLEC-2 on the platelet surface is reduced after platelet stimulation with ADP or podoplanin. Interestingly, the aggregation response was enhanced in the presence of podoplanin when platelets were co-stimulated with arachidonic acid, ADP, or collagen. Using these results, the authors proposed a model in which CLEC-2 is internalized during platelet activation, and sCLEC-2 is taken up by activated platelets in cancer patients. In contrast to previous studies, the authors suggest that CLEC-2 is not cleaved (shed) from the platelet surface and released into the blood plasma in cancer patients.

The authors showed that platelet surface expression of CLEC-2 is decreased upon platelet activation. CLEC-2 receptor shedding was excluded because sCLEC-2 levels were reduced in cancer patients. It is important to note that podoplanin expression is enhanced in many cancers, including CRC, lung squamous cell carcinomas, cervical cancer, and oral cancer. Cancer-associated fibroblasts can also express podoplanin, which is correlated with cancer malignancy and poor prognosis in lung, breast, pancreatic, and liver cancer. Podoplanin on the surface of cancer cells induces platelet aggregation, facilitating hematogenous cancer metastasis. Thromboinflammation can also induce ectopic podoplanin expression in vascular endothelial cells or macrophages, which may also contribute to cancer-associated thrombosis. Is it possible that platelet CLEC-2 is shed, and sCLEC-2 is trapped in the microenvironment of primary or metastatic tumors, leading to reduced sCLEC-2 levels in the blood plasma of cancer patients? This trapping/uptake mechanism probably does not exist in glioblastoma; therefore, sCLEC-2 levels are increased in the blood plasma of patients with glioblastoma. The authors could conduct experiments to investigate this hypothesis. This possibility can be addressed in vitro using different cancer cell lines, such as breast, colon, and glioblastoma, by testing their ability to uptake or bind sCLEC-2. Additionally, patient tumors could be lysed and tested for the levels of CLEC-2 and sCLEC-2.

Are the studied cancer patients still alive? Do they receive any medication or treatments? It would be useful to include this information and correlate sCLEC-2 levels after treatment with overall survival.

Wang et al. demonstrated that gastric cancer cell-resident CLEC-2 prevents AKT activation and glycogen synthase kinase-3 beta signaling, reducing invasiveness and expression of epithelial-to-mesenchymal transition markers in gastric cancer cell lines (Wang et al., 2016, Gastroenterology). CLEC-2 also inhibits the expression of PI3K subunits in a SYK-dependent manner. These findings indicate a tumor-inhibitory role for cancer cell-resident CLEC-2. Notably, these experiments were conducted under normal platelet count conditions in mice, suggesting that platelet-derived CLEC-2 is dispensable in this cancer type. This study revealed a decrease in CLEC-2 expression in gastric cancer tissues and cell lines, which can be cited and discussed here.

Several single nucleotide variants (SNVs) were identified in the promoter, 5´UTR, and coding regions of the CLEC-2 gene (CLEC1B). Notably, homozygous haplotype group 3 exhibited significantly lower sCLEC-2 levels in the German population. According to allele frequencies, most of the SNVs showed significant differences between European and Asian populations. Therefore, it was proposed that sCLEC-2 plasma levels are genetically linked and may vary among individuals and between Asian and European populations. Is it possible that the decreased sCLEC-2 levels in patients with CRC could be attributed to haplotype variability, which differs from the Asian population where sCLEC-2 levels are increased in patients with CRC?

Author Response

The authors showed that platelet surface expression of CLEC-2 is decreased upon platelet activation. CLEC-2 receptor shedding was excluded because sCLEC-2 levels were reduced in cancer patients. It is important to note that podoplanin expression is enhanced in many cancers, including CRC, lung squamous cell carcinomas, cervical cancer, and oral cancer. Cancer-associated fibroblasts can also express podoplanin, which is correlated with cancer malignancy and poor prognosis in lung, breast, pancreatic, and liver cancer. Podoplanin on the surface of cancer cells induces platelet aggregation, facilitating hematogenous cancer metastasis. Thromboinflammation can also induce ectopic podoplanin expression in vascular endothelial cells or macrophages, which may also contribute to cancer-associated thrombosis. Is it possible that platelet CLEC-2 is shed, and sCLEC-2 is trapped in the microenvironment of primary or metastatic tumors, leading to reduced sCLEC-2 levels in the blood plasma of cancer patients? This trapping/uptake mechanism probably does not exist in glioblastoma; therefore, sCLEC-2 levels are increased in the blood plasma of patients with glioblastoma. The authors could conduct experiments to investigate this hypothesis. This possibility can be addressed in vitro using different cancer cell lines, such as breast, colon, and glioblastoma, by testing their ability to uptake or bind sCLEC-2. Additionally, patient tumors could be lysed and tested for the levels of CLEC-2 and sCLEC-2.

This interesting hypothesis is included in the Discussion section:

Binding of sCLEC-2 to PDPN on CTCs should also be considered. In this concept, platelet CLEC-2 is shed and sCLEC-2 is trapped in the microenvironment of primary or metastatic tumors leading to reduced sCLEC-2 levels in the blood plasma of cancer patients. The trapping mechanism, probably, does not exist in glioblastoma. In a follow-up study, we will address the mechanism by using different cancer cell lines including CRC, BC and glioblastoma to test the binding or uptake of sCLEC-2.

Are the studied cancer patients still alive? Do they receive any medication or treatments? It would be useful to include this information and correlate sCLEC-2 levels after treatment with overall survival.

Unfortunately, data about the treatment or outcome of patients is not available to us because this is not covered by the ethical vote of our study.

Wang et al. demonstrated that gastric cancer cell-resident CLEC-2 prevents AKT activation and glycogen synthase kinase-3 beta signaling, reducing invasiveness and expression of epithelial-to-mesenchymal transition markers in gastric cancer cell lines (Wang et al., 2016, Gastroenterology). CLEC-2 also inhibits the expression of PI3K subunits in a SYK-dependent manner. These findings indicate a tumor-inhibitory role for cancer cell-resident CLEC-2. Notably, these experiments were conducted under normal platelet count conditions in mice, suggesting that platelet-derived CLEC-2 is dispensable in this cancer type. This study revealed a decrease in CLEC-2 expression in gastric cancer tissues and cell lines, which can be cited and discussed here.

This important point is added to the Discussion and the new Reference 29 is included:

As shown in a previous study, the decrease in CLEC-2 expression in gastric cancer tissues and cell lines led to lower invasiveness of the cells [29]. These findings indicated a tumor-inhibitory role for cancer cell-resident CLEC-2. Notably, these experiments were conducted under normal platelet count conditions in mice, suggesting that platelet-derived CLEC-2 is dispensable in this cancer type.

Several single nucleotide variants (SNVs) were identified in the promoter, 5´UTR, and coding regions of the CLEC-2 gene (CLEC1B). Notably, homozygous haplotype group 3 exhibited significantly lower sCLEC-2 levels in the German population. According to allele frequencies, most of the SNVs showed significant differences between European and Asian populations. Therefore, it was proposed that sCLEC-2 plasma levels are genetically linked and may vary among individuals and between Asian and European populations. Is it possible that the decreased sCLEC-2 levels in patients with CRC could be attributed to haplotype variability, which differs from the Asian population where sCLEC-2 levels are increased in patients with CRC?

Unfortunately, genetic data of the patients were not available to us because this is not covered by the ethical vote of our study. Nevertheless, we included a paragraph in the Discussion in order to address this point:

Another aspect regarding differences between HC and patients is given by the individual genetic background and ethnicity. However, genetic data of the cancer patients in the pre-sent study were not available but we assume that the ethnicity of the HC and the cancer patients is very similar. In a previous study, we reported a correlation of haplotypes of the CLEC1B gene encoding CLEC-2 with the sCLEC-2 level in healthy individuals [25]. The haplotypes were defined by 7 single nucleotide variations (SNVs: rs10505743, rs11053538, rs4764178, rs76016091, rs2273986, rs2273987, rs521040) and the most frequent haplotype was associated with significantly lower sCLEC-2 levels. According to the gnomAD data-base (https://gnomad.broadinstitute.org/), the minor allele frequency of some SNVs is dif-ferent in the European, African and Asian population. E.g., rs4764178 and rs2273986 are more frequent in Europeans than in Asians, whereas, rs2273987 and rs521040 are more frequent in Asians than in Europeans.

Round 2

Reviewer 1 Report

Comments and Suggestions for Authors

The revisions are OK for me and agree with the publication of this manuscript.

Reviewer 2 Report

Comments and Suggestions for Authors

No comments